# Epidemiologic Aspects of Mycetoma in Africa

**DOI:** 10.3390/jof8121258

**Published:** 2022-11-29

**Authors:** Michel Develoux

**Affiliations:** Laboratoire de Parasitologie-Mycologie, Centre Hospitalier Universitaire Saint-Antoine, Assistance Publique-Hôpitaux de Paris, 184 rue du Faubourg Saint-Antoine, 75012 Paris, France; micheldeveloux@yahoo.fr

**Keywords:** mycetoma, Africa, epidemiology, geographical distribution, etiological agents, mode of infection, prevalence

## Abstract

Mycetoma is a chronic, disabling infection caused by fungi or actinomycetes that affects the disadvantaged rural populations of arid tropical regions. The identification of etiological agents is long, difficult, and often imprecise or unsuccessful. Recently developed molecular methods can be used to identify causal agents at the species level. However, diagnosis can only be implemented in specialized laboratories. For these reasons, the distribution of causal agents in endemic African countries remains approximate. It is known that the pathogenic organisms of mycetoma are present in the environment, introduced as a result of injuries or trauma. There are still unknowns concerning the natural habitats of agents and the mode of infection. A potential association between mycetoma and acacia was uncovered in Sudan, allowing the elaboration of a risk map of the country. A new hypothesis for the mode of contamination involves the intervention of an intermediate host. The first surveys in Sudanese endemic villages gave a higher prevalence than the previous estimates, indicating that the prevalence of mycetoma in endemic African countries has previously been underestimated.

## 1. Introduction

Mycetoma is defined as all processes in which fungal or actinomycotic agents of exogenous origins produce grains. This chronic infection occurs in the rural populations of arid tropical zones. It is a disease of poverty that is destructive and disabling. Africa is the most affected continent; of the 19,494 listed cases, 10,608 are from Sudan [1]. Clinical aspects of eumycetoma (due to fungus) and actinomycetoma (due to actinomycetes) are similar, and these sources represent the main origins of the disease. The medical treatment of mycetoma differs according to the causative agents, for instance, actinomycetoma patients respond well to antibiotic treatment, and eumycetoma patients to antifungal treatment [2]. In 2016, mycetoma was added to the WHO list of neglected tropical diseases [3]. This contributed to a renewed interest in this disease, with an increase in studies and publications. Nevertheless, details of some of its epidemiological aspects (ecological niche of etiologic agents, mode of contamination, geographical distribution, prevalence) remain unclear.

## 2. Etiological Mycetoma Agents in Africa

In the endemic mycetoma zones of Africa, the etiological agents are dominated by *Madurella mycetomatis*, *Actinomadura madurae*, and *Streptomyces somaliensis* [4]. Molecular methods have been applied in the identification of black-grain mycetoma agents [5]. Before this, identification at the species level using classical mycological methods was difficult and sometimes impossible [6]. Cultures are often negative or contaminated, and most species are nonsporulating. The analysis of various *M. mycetomatis* strains revealed the possibility of other *Madurella* species. In 2012, De Hoog and associates described *Madurella fahalii*, a new species causative of black-grain mycetoma in Sudan [7]. It is closely related to *M. mycetomatis* and *M. pseudomycetomatis* that were first identified in China. An isolate from Indonesia was described as *Madurella tropicana*. Recently, molecular identification of *Madurella* sp. strains using direct PCR confirmed the predominance of *M. mycetomatis* (90.1%) in Sudan and the rarity of *M. fahalii* [8], and the first report of *M. tropicana* was as a mycetoma agent in Africa. One sample matched with *Sphaerulina rhododetricola*, a phytogenic species, and molecular methods identified new fungal species implicated in African eumycetoma: *Pleurostomophora ochracea* [9], *Curvularia pseudolunata* [10], *Chaetomium atrobrunneum* [11], and *Microascus gracilis* [12].

In 2008, nine strains of actinomycetes from Sudan, identified as *Streptomyces somaliensis*, were the subject of a polyphasic taxonomic study [13]. A subclade appeared as a genomic species with identical phenotypic profile in five strains. It was recognized as a new species: *Streptomyces sudanensis*. The real location and importance of *S. sudanensis* in the African mycetoma zone must be specified. *Actinomadura mexicana*, identified initially in Latin America, was shown to also be a causative agent of actinomycetoma in Sudan [14]. Finally, *Actinomadura bengladeshensis* was identified for the first time in France from a patient with mycetoma, and it had originated from Mali [15].

The newly identified causative agents of mycetoma in Africa during the 2002–2021 period are grouped in Table 1.

## 3. Agents of Mycetoma in the Environment

There are few studies aimed at determining the presence of causative mycetoma agents in the environment in Africa and other endemic zones. On the Senegal riverbanks, an endemic area for black-grain mycetoma, *Falcimosporma senegalensis* (ex-*Leptosphaeria senegalensis*) was cultivated from dry thorns periodically covered by mud during annual floods [16]. *Neotestudina rosati*, an exceptional fungal agent of white-grain mycetoma, was isolated twice from dry, sandy soils. *Madurella* sp., the most frequently identified causative agent implicated in mycetoma in the populations living of the banks of Senegal river, could not be cultivated from soil or thorns. This study revealed the frequency of some agents of eumycetoma in Senegal and the probable existence of a specific biotope for each species. Ahmed and associates in Sudan [17] also failed to identify *Madurella* sp. in fungal colonies isolated from soil and thorns. However, the presence of *Madurella mycetomatis* was detected in the same environment using molecular methods. A total of 17 of the 74 (23%) soil samples were positive for *M. mycetomatis* DNA, and only 1 out of 22 (5%) of the thorns. These data support the acquisition of *M. mycetomatis* from the environment. The detection of various mycetoma agents was then performed using fungal metabarcoding analysis of soil DNA in an endemic zone of Sudan [18]. Twelve mycetoma causative fungal agents were found to be especially prevalent, among others: *M. mycetomatis*, *M. fahalii*, *F. senegalensis*, *F. tompkinsii*, *Fusarium solani*, and *Curvularia lunata*. These results demonstrate the diffusion of various causative agents in the soils of endemic mycetoma areas and the applicability of fungal metabarcording analysis in geographical mapping. Another study, also performed in Sudan, used a phylogenic approach to specify the natural habitat of *M. mycetomatis* [19]. This species appeared nested within the Chaetamiaceae family, frequently isolated from animal dung and enriched soils. Several hypotheses can be raised regarding a new habitat for *M. mycetomatis* and its mode of transmission, which could lead to the development and application of preventive measures.

## 4. Influence of Climate and Environmental Factors in the African Endemic Zone for Mycetoma

The close relationships between climate conditions and the localization and limits of the African endemic zone for mycetoma has long been known. In 1961, from a study of more than 200 patients from Senegal–Mauritania, the distribution of etiological agents could be distinguished according to three quite distinct zones [20]: First, the northern part is the Atar region in the desert of Mauritania, with annual rainfall of 50–250 mm and a predominance of *S. somaliensis* and a minority of *Madurella* sp. and *Falcimosporma* sp. Second, the intermediate zone, with 250–500 mm annual rainfall, is situated along the Senegal river, a natural frontier between Senegal and Mauritania. Most of the species identified were *M. mycetomatis* and *F. senegalensis*, with some cases resulting from *A. madurae* and *A. pelletieri*. Thirdly, the southern zone, with 500–800 mm annual rainfall, is the domain of red-grain mycetoma due to *A. pelletieri*, with some fungal white-grain mycetoma. These results were confirmed by further studies in Senegal [21,22].

The area situated between 30° N and 15° S characterizes the endemic zone of Africa. The most affected regions have an annual rainfall between 50 and 1000 mm, including from the west to the east of the continent: Mauritania, Senegal, Mali, Niger, Chad, Sudan, Ethiopia, Republic of Djibouti, and Somalia. It can be represented by a band dubbed “the mycetoma belt” [1]. The reported distribution of the main species in Mauritania–Senegal reveal it is partially observed in other countries of the mycetoma belt. *Actinomadura pelletieri* represents 28% and 32% of the species identified in Mali [23] and Niger [24], respectively, but as an etiological agent, it has become a rare in the central and east part of the mycetoma belt. *F. senegalensis* has a regional distribution and is mainly found in the Senegal river banks area. 

The potential risk of mycetoma infection in Sudan and South Sudan has been evaluated using niche modeling (ENM). It has been noted for decades that acacia trees represent most of the flora in the endemic mycetoma zone of Senegal and Sudan. Samy and associates [25] integrated mycetoma case records of the Mycetoma Research Center (MRC) of Khartoum and geospatial data (land surface, temperature, soil characteristics, and greenness). ENM based on environmental predictors indicates that greater risks are associated with the east–west belt across Sudan. Comparing niches of mycetoma and acacia trees, significant similarity was found. These results represent a step toward the development of valid risk maps for the infection. 

The study of Samy and associates integrated only 44 proven cases of mycetoma. A later study involved 6983 patients (5513 eumycetoma and 1470 actinomycetoma) diagnosed at the MRC during 1991–2018 [26]. Environmental predictors included precipitation, temperature, soil composition and pH, livestock distribution, proximity to water sources, elevation and related topographical variables, and distribution of thorny vegetation. The distribution results showed, as in previous studies, that infection is found in a belt characterized by an annual rainfall of 50–1000 mm and a high temperature. The coldest quarter of the year, with 20–25 °C temperature, seems most favorable for actinomycetoma. Daily variation of temperature appears to be an important predictor of eumycetoma. Fungal agents of mycetoma cannot survive when the daily temperature exceeds 15 °C. Most of the patients were found to live close to water sources and rivers, which could more reflect the behaviors of these populations rather than environmental factors. For the first time, the mineral composition of soils was studied, with the finding that low soil calcium and sodium concentrations are suitable for infection. A greater number of species of acacia were predictors of eumycetoma. Ganawa and associates [27] found, in a highly endemic state of Sudan, an association between geographical distribution of patients and soil type. The patient domiciles had light clay soil in the majority of cases (80%), followed by 13% with sandy loam soil. Most of the localities had the same cover and vegetation. In northeast Mexico, where actinomycetoma are predominant, cases occurred in places with a mean annual temperature ranging between 17.2 and 25 °C and a mean precipitation between 375.5 and 2713.3 mm. Most cases were reported in areas with kastanozem and lithosol soil types [28]. This study confirms the effect of soil type and climatic conditions on mycetoma case distribution.

## 5. Mode of Contamination

The exact mode of contamination in the case of mycetoma remains unclear. Etiological agents have been found in the environment and are supposedly introduced as a result of trauma. Affected populations (farmers, breeders, and housewives) live in rural areas and are submitted to various traumas or mini-traumas because of their occupations. An incident involving trauma was recalled by 20% of patients in a cohort of 6792 [29]. The percentage could be underestimated, as some trauma could have been forgotten about or ignored in the case of mini-trauma. An indirect argument in favor of the role of trauma is the predominance of the lesions at the extremities, mainly the foot (Figure 1). In the Sudanese records of the MRC, the foot was the most common localization (76%) and the hand the second (7.5%). Various traumas have been implicated: thorns picks, splinter cuts, injury with sea urchin spine or fish bone, public road accident, trauma with tools, traumas inflicted during sport activities, and animal bites, among others. Thorns or plant residues are sometimes found in resection specimens (Figure 2), and this is another argument for its role in the introduction of mycetoma agents in human.

A study comparing four *Madurella* species and *Chaetonium*, *Chaetomidium*, *Thiclavia*, and *Papulaspora* confirmed that *Madurella* species are, phylogenetically, a member of the Chaetomiaceae family. These fungi are often isolated from dung and enriched soils. The abundance of dung in the proximity of habitations has been noted in endemic Sudanese villages [30]. Various animals, particularly ruminants, lives closely to houses. The animal enclosures are built with acacia tree branches. All the conditions are met such that inhabitants are more exposed to thorn tricks, which have been implicated in the transmission of eumycetoma. Most are barefoot and have poor access to hygiene facilities, which increases the risk and consequences of trauma with thorns. Nevertheless, numerous authors consider that thorn pricks cannot represent the only mode of contamination, and new hypotheses have been raised in which there is involvement of an intermediate host [31]. For example, ticks could have a role in the transmission of *M. mycetomatis* [32]. Some observations favor the intervention of ticks: the high prevalence of people and animals bitten by ticks in an endemic village compared with a non-endemic one, the documented presence of *M. mycetomatis* DNA in a pool of ticks. However, further studies are needed to confirm these preliminary results. In endemic areas, numerous inhabitants are subject to risk of contamination, but few will develop infection. This is another unclear aspect of the disease that may indicate the involvement of immunologic and genetic predispositions [33].

## 6. Geographical Distribution of Mycetoma in Africa

The first case of mycetoma in Africa was reported in Saint-Louis by Le Dantec in 1894 (Senegal), decades after the mycetoma belt was described. Most of the studies on this disease are from two countries included in the belt: Sudan and Senegal. Data are still lacking on the existence and the importance of the disease in other countries of the mycetoma belt. Since the infection was added to the WHO list of neglected tropical disease, 198 cases have been reported in Senegal [22], 87 cases in Mauritania [34], and 19 cases in Mali [35]. All three found a predominance of eumycetoma due to black grains. The results remain imprecise because of the significant percentage of indeterminate type and the lack of identification of the causal agent in most cases. There are no recent studies concerning Niger, with the first published in 1988 [24], recording the predominance of actinomycetoma. Northern Nigeria is included in the mycetoma belt with an annual rainfall of 600–1000 mm, but only a few older publications are available [36]. In 1970, Destombes et al. [37] published the results of deep mycoses diagnosed by histopathology during a twenty-year period (1947–1968) in the institute Pasteur of Brazzaville (Republic of Congo). There were 95 mycetoma patients, 81 of which were from Chad and north Cameroon, part of a zone with warm semi-arid climate. There was a slight predominance of actinomycetoma at 52% versus 48% of eumycetoma. Species, identified based on histologic criteria, were *Madurella mycetomatis*: 35; *Madurella grisea*: 1; *Leptosphaeria senegalensis:* 3; *Cephalosporium* sp.: 1; *Actinomadura pelletieri*: 20; *Streptomyces madurae*: 10; and *Streptomyces somaliensis*: 1. L’Escalopier and associates published a retrospective study of 132 cases who had surgery in N’Djamena, Chad, from 2007 to 2018 [38]. The authors were surgeons deployed within the French military assistance. Unfortunately, diagnosis was only clinical; nevertheless, this report demonstrates the persistence and severity of the disease in Chad. Because of the presence of the MRC, established in Khartoum in 1991, the most detailed reports are from Sudan [29]. Of 6792 patients, 3177 (47%) had fine needle aspiration (FNA) for cytology. The diagnosis was *M. mycetomatis:* 2379 (75%); *A. madurae*: 316 (10%); and *S. somaliensis*: 277 (7%). *A pelletieri* represented only 39 cases (1%). Results concerning countries of the horn of Africa included in the mycetoma belt (Eritrea, Ethiopia, Republic of Djibouti, Somalia) are few or very dated. A total of 50 cases were reported from the Republic of Djibouti (the former "Cote française des Somalis" ) in 1958 [39]. Causal agents were actinomycetes in 28 cases (*S. somaliensis*: 23; *A. madurae*: 4; and *Nocardia* sp.: 1) or fungi (*Madurella* sp.: 20; *Falcimospora* sp.: 1; and *P. romeroi*/*M. grisea*: 1). A survey conducted from 1959 to 1964 in Somalia included 103 observations, with 94 pathogens identified: 60 fungi (*Madurella* sp.: 44, *F. senegalensis*: 1, *Fusarium* sp.: 1, *Zopfia rosatii*: 3; and unidentified: 1) and 34 actinomycetes (*S. somaliensis*: 24; *A. madurae*: 4; *A. pelletieri*: 3; and *Nocardia* sp.: 3) [40]. During the first twenty years of this century, some mycetoma were reported from Ethiopia [41,42]. Rare, imported cases from Europe [43,44,45] and Israel [46] concerned patients native to the horn of Africa. There are still unexplored foci in the endemic African zone. In the hospital of Tanguieta, northern Benin, near the frontier with Burkina Faso and Niger, mycetoma are regularly observed in patients native to Benin, Burkina Faso, or Niger (Dr Priuli, personal communication). No cases from Benin have ever been published, and very few from Burkina Faso, a country in the Sahelian zone with a Sudano-Sahelian climate in more than half of the country.

Data on mycetoma are still lacking, inadequate, or obsolete in the mycetoma belt. The available results indicate that eumycetoma, essentially due to black grain, is more frequent than actinomycetoma, representing a not insignificant proportion: 36.8% in Senegal [22] and 30% in Sudan [29]. *Madurella* sp., *A. madurae*, and *S. somaliensis* are the main causal agents. It is important to have new studies on the importance and distribution of mycetoma in the endemic zone, with accurate identification of species. This supposes the utilization of molecular techniques. It must be pointed out that characteristics of the endemic mycetoma zone have varied over time, due essentially to climatic change. The lower line of the belt represented by the 1000 mm isohyet has been modified, moving to the south. An increase in mycetoma cases was noted in areas where the disease was previously exceptional until now [21].

Outside the mycetoma belt, in sub-Saharan Africa, there are recent results from some countries. In Ivory Coast, during a twenty-year period, 87 cases were diagnosed in the laboratories of Abidjan university [47]. Globally, fungi were slightly more frequent (52.9%) than actinomycetes (47.1%). In the forest, actinomycetoma is predominant, whereas eumycetoma is more common in the savanna. In the north, *M. mycetomatis* is the major species, and in the forest zone, the major species are *Nocardia brasiliensis* and *S. apiosermum*/*P. boydii*. In a study of 61 cases in Togo, a coastal country close to Ivory Coast, the predominance of eumycetoma was more marked [48]. The differences in the results between the different climate zones were also noticed. All the eumycetoma were observed in the savannah and the actinomycetoma in more humid regions of Togo. In the Democratic Republic of Congo, equatorial Africa, the main species isolated from the rare cases of mycetoma were found to be *Nocardia brasiliensis* and *S. apiospermum*/*P. boydii* [49]. These results are characteristic of mycetoma agents in humid African areas.

Some data are available from east Africa, Kenya, Uganda, and Tanzania. In Uganda, the estimated number of people living with the disease is 3683 [50]. The majority of cases are eumycetoma (89%), but no identification of genus or species was given. *Nocardia* sp. (5%) and actinomycetes (4%) represent the other causal agents.

In Austral Africa, most of the rare notifications are from South Africa [51].

Madagascar is the focus of chromoblastomycosis in the world. Mycetoma, less common, was reported in the south of the island, an area with semi-arid climate with a flora consisting of thorny shrubs [52].

In North Africa, Morocco, Algeria, and Tunisia represent a hypoendemic focal point that has long been known of [53,54,55]. *A. madurae*, the most frequent agent, was first described in Algeria in 1894. Eumycetoma are a little rarer than actinomycetoma, mostly due to *M. mycetomatis*. Egypt is a low-endemic country but the incidence is higher than previously reported [56].

## 7. Prevalence and Incidence

Until recently, the results on mycetoma came from university hospitals of Africa, located far from the foci. More recently, results were published from regional hospitals that are now more equipped [35]. Hospitalized mycetoma represents the most severe cases. The statistics do not usually include patients with early or slowly progressive lesions, who are not consulted at this stage of the disease. It is therefore difficult to obtain valid data, and the prevalence is probably underestimated. The average prevalence of mycetoma cases in the review of van de Sande [4] was obtained from the number of cases reported in a year in a country divided by the total population of that country of that same year (Figure 3). The most elevated prevalence was found in Mauritania: 3.49/100,000 inhabitants and 1.81/100,000 inhabitants in Sudan. The annual numbers of cases per year were 69.7 and 106, respectively. In an endemic village of Sudan, the prevalence was found to be 14.5/1000, much higher than that previously published for the country [30]. It was the first field study on mycetoma, and all inhabitants were examined. An elevated prevalence was also found in eastern Sudan [57]. In sixty villages, 41,176 subjects were surveyed, of which mycetoma were identified in 359, corresponding to an overall prevalence of 8.3/1000. The prevalence map revealed that patients were clustered in the central and northeastern part of the locality. The local environment and sanitary conditions could explain this distribution.

## 8. Host of Mycetoma

The predominance of mycetoma in males has been known for a long time. In the review of van de Sande, most of the cases were found in males: 4060 versus 1175 in females [4]. This difference cannot be explained by a difference of exposition to the risk of contamination according to sex. A possible role of hormonal influence in mycetoma susceptibility is suspected. A difference in hormone level was noted between mycetoma patients and healthy patients [58], but in vitro *M. mycetomatis* growth was not found to be affected by hormone level. Most of the patients are aged between 11 and 40: 70% of 5240 cases [4]. In a case–control study conducted in villages from Sennar state, eastern Sudan, there were nearly three times more cases in the 16–30 years age group compared with the ≤15 years age group [59]. This is the first population-based case–control study of sociodemographic risk factors for mycetoma. One of the risk factors identified was a history of local trauma. In previous studies, only a minority of patients had such a medical history. Other factors found to contribute to the odds of mycetoma are being unmarried and owning livestock. The risk factor found in this study could serve in diagnosis of mycetoma and be used in a control program.

## 9. Conclusions

Recent progress has been made in the knowledge of mycetoma epidemiology; however, some aspects remain unclear or poorly understood. The distribution and nature of causal agents inside the mycetoma belt, endemic African area, have been insufficiently studied. It is therefore essential to have more data about the species involved and their distribution in the different countries of the belt. A new hypothesis on the mode of transmission of the infection has been raised and is still the subject of research. Environmental factors to mycetoma infection were identified and have led to the elaboration of risk maps for the disease. One of the main priorities is to obtain reliable prevalence and incidence data for endemic Africa. The first results with field studies were obtained. All these new data have been obtained from studies conducted in Sudan, and similar studies should be conducted in other countries of the mycetoma belt.

## Figures and Tables

**Figure 1 jof-08-01258-f001:**
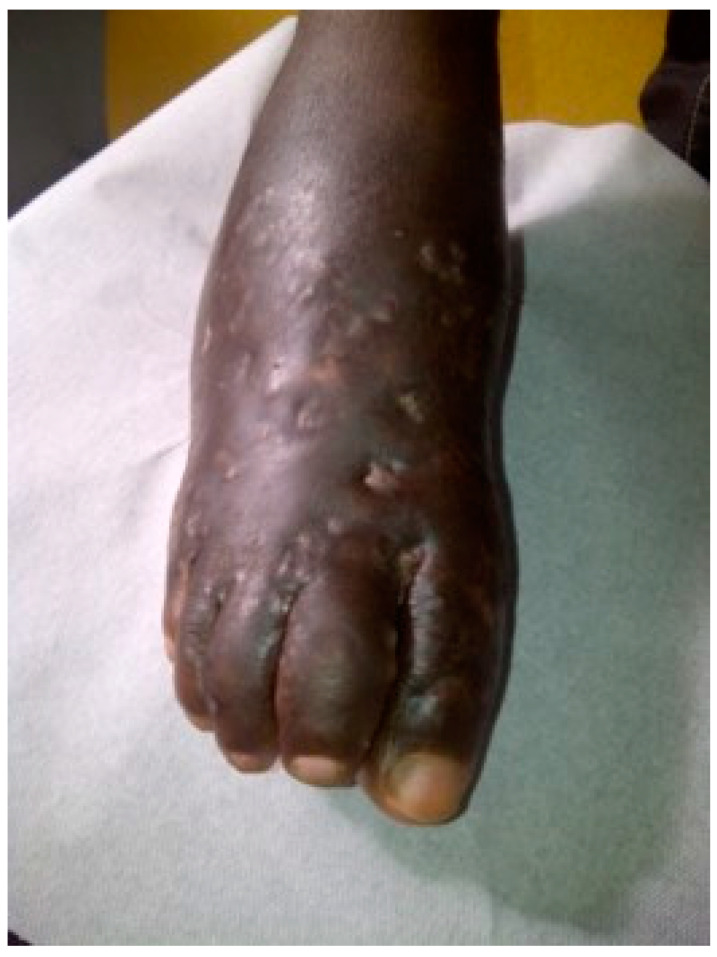
Black-grain eumycetoma in a Senegalese patient.

**Figure 2 jof-08-01258-f002:**
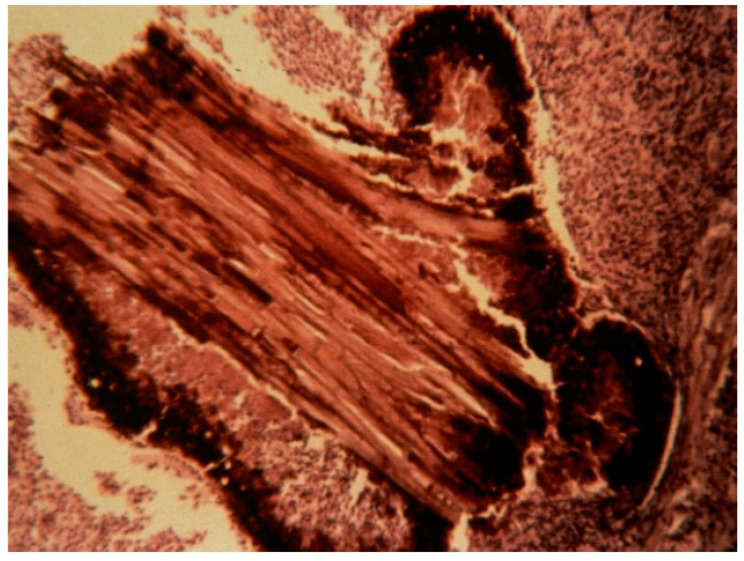
Thorn in a *Falcimosporma* sp. grain (R. Camain, Dakar).

**Figure 3 jof-08-01258-f003:**
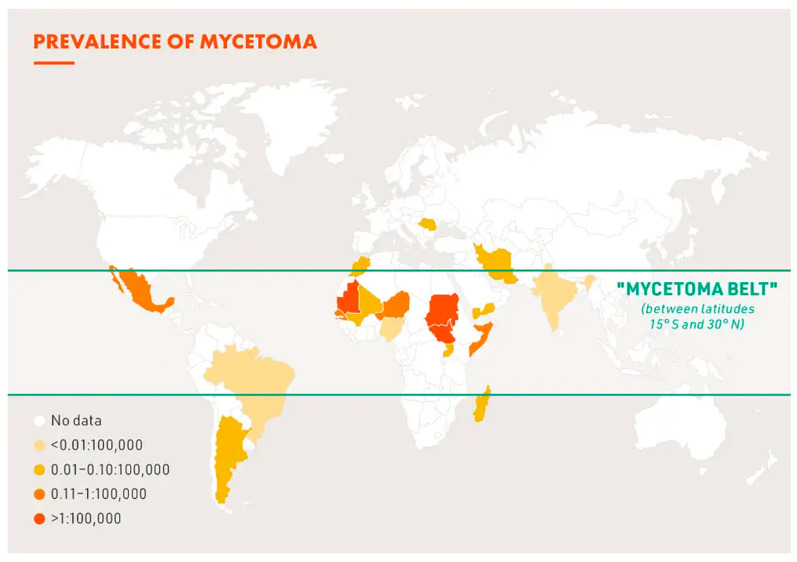
Prevalence of mycetoma (DNDi 2019 adapted from van de Sande WWJ [4]).

**Table 1 jof-08-01258-t001:** Newly identified causative agents of mycetoma in Africa (2002–2021).

Eumycetoma	
**Black grain**	
*Madurella fahalii*	[5]
*Madurella tropicana*	[8]
*Curvularia pseudolunata*	[10]
*Sphaerulina rhododendricola*	[8]
*Chaetomium atrobrunneum*	[11]
**White grain**	
*Microascus gracilis*	[12]
**Yellow grain**	
*Pleurostomophora ochracea*	[9]
Actinomycetoma	
**White or yellow grain**	
*Streptomyces sudanensis*	[13]
*Actinomadura bangladeshensis*	[15]
*Actinomadura mexicana*	[14]

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
