# Peer review of "Epidemiologic Aspects of Mycetoma in Africa"

_jof, 2022, doi:10.3390/jof8121258_

Round 1

Reviewer 1 Report

Dear Author,

The paper is really important in the field; however it require some modifications please see my comments in the attached PDF file 

Author Response

Dear reviewer i made some revision of the english language as your request

regards m develoux

Reviewer 2 Report

The present manuscript is a review on the situation of mycetoma in African countries. It is well written and summarizes the knowledge about the infection in the African continent. It is well written, with few errors. My suggestions are the following: 

1. Table 1, put the names of the microorganisms in italics

2. Page 3, Line 115: Please provide the name of the Acacia trees

3. Please discus the following paper in chapter 4 (Influence of climate..)(even and it is not in Africa): Cardenas-de la Garza JA, Welsh O, Cuellar-Barboza A, Suarez-Sanchez KP, Cruz-Gomez LG, De la Cruz-Valadez E, Ocampo-Candiani J, Vera-Cabrera L. Climate, soil type, and geographic distribution of actinomycetoma cases in Northeast Mexico: A cross-sectional study. PLoS One. 2020 May 8;15(5):e0232556. doi: 10.1371/journal.pone.0232556. 

4. Fig 2. The picture is of low quality. Please change it for one with better resolution

Author Response

Dear reviewer,

1. I put the name of the microorganisms in italic.

2. I was unable to provide the name of the acacia because it was not specified in the reference.

3. I discussed the paper of cardenas-de-la Garza and associated.

4. I proposed for Figure 2 a picture of better resolution

Reviewer 3 Report

Good effort, author has worked hard to gather the information. needs to improve expression and language.

Author Response

Dear reviewer

I tried to improve the English language of the submitted paper regards

Round 2
